# AutoGDA: Automated Graph Data Augmentation for Node Classification

**Tong Zhao♠♥, Xianfeng Tang♣, Danqing Zhang♣, Haoming Jiang♣, Nikhil Rao♣, Yiwei Song♣, Pallav Agrawal♣, Karthik Subbian♣, Bing Yin♣, Meng Jiang♠**

♠ University of Notre Dame, Notre Dame, IN, USA
♥ Snap Inc., Seattle, WA, USA
♣ Amazon.com Inc., Palo Alto, CA, USA

## Abstract

Graph data augmentation has been used to improve generalizability of graph machine learning. However, by only applying fixed augmentation operations on entire graphs, existing methods overlook the unique characteristics of communities which naturally exist in the graphs. For example, different communities can have various degree distributions and homophily ratios. Ignoring such discrepancy with unified augmentation strategies on the entire graph could lead to sub-optimal performance for graph data augmentation methods. In this paper, we study a novel problem of automated graph data augmentation for node classification from the localized perspective of communities. We formulate it as a bilevel optimization problem: finding a set of augmentation strategies for each community, which maximizes the performance of graph neural networks on node classification. As the bilevel optimization is hard to solve directly and the search space for community-customized augmentations strategy is huge, we propose a reinforcement learning framework AutoGDA that learns the local-optimal augmentation strategy for each community sequentially. Our proposed approach outperforms established and popular baselines on public node classification benchmarks as well as real industry e-commerce networks by up to +12.5% accuracy.

## 1 Introduction

Data augmentation methods are widely used to improve the generalizability and robustness of machine learning (ML) models [1]. They aim to create plausible variations of existing data without the need of additional human efforts. It has been proved that customized data augmentation, i.e., customizing augmentation strategies for each (batch of) object, are beneficial for ML models [2–4]. For example, customized augmentation strategies [2, 5] have shown improved performance over having a uniform augmentation on the entire dataset. To this end, automated data augmentation methods efficiently seek the optimal customized augmentation strategies for samples/batches [2, 4–6].

Recently, with graph neural networks (GNNs) [7–10] emerging as one of the preferred approaches for learning on graph structured data, graph data augmentation methods [11–17] have shown promising results in improving GNNs. For example, DropEdge [18] randomly removes a fraction of edges in each training epoch to promote GNN's robustness during test-time inference. The AdaEdge [11] approach iteratively adds (or removes) edges between nodes that are predicted to have the same (or different) labels with high confidence. GAugM and GAugO [13] manipulate the graph structure according to edge probabilities learned by link predictors. Despite the promising improvements on various node classification tasks, existing graph data augmentation approaches *are manually designed for the entire graph* and *only explore graph properties and characteristics globally*.

---

*This work was done during the first author's internship at Amazon.com. Correspondence to: Tong Zhao <tzhao2@nd.edu;tzhao@snap.com>.

T. Zhao et al., AutoGDA: Automated Graph Data Augmentation for Node Classification. *Proceedings of the First Learning on Graphs Conference (LoG 2022)*, PMLR 198, Virtual Event, December 9–12, 2022.

It is more involved to apply automated data augmentation on graphs compared to images and text, because of the unique properties of graph data bring a great challenge to the effort. While existing automated augmentation approaches [2, 5] assume that samples are independent and identically distributed (i.i.d.) in the dataset, nodes in the graph are naturally connected and are dependent on each other in a non-Euclidean manner. Therefore, it is not straightforward to apply existing automated augmentation methods for graph data. On the other side, the unique properties of graph data may give us some clue to design new and effective solutions. Nodes in the graph are naturally grouped into communities [19, 20], providing a natural separation of data objects (nodes) for node classification. Chiang et al. [21] show that nodes from the same community are the most important neighbors for aggregation-based graph learning algorithms. As communities in graphs such as social networks are usually disparate in characteristics [22–24] such as density, centrality, homophily, etc., we argue that data augmentation strategies should be localized (community-specific) to achieve optimal results. However, how to augment graph data according to the localized characteristics of communities in the graph remains underexplored.

To address the aforementioned challenges, we propose to tackle down the problem of automated graph data augmentation from the local perspective, i.e., communities in graphs. We first analyze the disparate characteristics of communities using benchmark datasets. Motivated by observations and insights, we define automated graph data augmentation as a bilevel optimization problem, that is, to learn the augmentation strategies that lead to the best node classification performance of GNNs. As finding the optimal augmentation strategies requests combinatorial optimization, it is impractical in real world due to huge computational cost. We propose AutoGDA that learns community-customized augmentation strategies with a reinforcement learning (RL) approach, inspired by the auto-augmentation literature in computer vision [2, 5]. Specifically, given communities in a graph, AutoGDA relies on an RL-agent to sequentially pick up the optimal strategy from several graph data augmentation operations for each community. The RL-agent in AutoGDA generalizes the learning and selection of augmentation method from one community to another, and thus automates and accelerates the process of finding localized augmentation strategies.

We conduct extensive experiments across different GNN backbones and datasets to evaluate AutoGDA against state-of-the-art baselines. We demonstrate that AutoGDA with traditional community detection algorithms (e.g., the Louvain method [25]) and existing graph data augmentation operations (DropEdge [18], GAugM [13], and AttrMask [26]) can achieve consistent performance improvements over the baselines. Specifically, AutoGDA shows up to 12.5% over the best-performed baseline method. Moreover, we show that the graph representations learned by AutoGDA are robust against graph adversarial attacks [27].

Our main contributions are as follows.

- We tackle down the problem of automated graph data augmentation for supervised node classification by proposing community-customized augmentations from a localized perspective. To the best of our knowledge, we are the first to investigate community-customized graph data augmentation for the task of node classification.

- We propose AutoGDA, an RL-based framework that automatically learns optimal community customized graph data augmentation strategies. The AutoGDA framework is flexible on the augmentation operations and can be easily generalized to heterogeneous graphs.

- We conduct extensive experiments on six benchmark datasets (including two real industrial graph anomaly detection benchmarks) with three widely used GNN backbones to validate AutoGDA. The experimental results show that (1) AutoGDA consistently outperforms state-of-the-art graph data augmentation baselines across all datasets, and (2) AutoGDA learns robust representations that give comparable or better classification performance than state-of-the-art graph defensing methods against adversarial attacks.

## 2 Preliminaries and Problem Definition

**Notations** Let $G = (\mathcal{V}, \mathcal{E})$ be an undirected graph of $N$ nodes, where $\mathcal{V} = \{v_1, v_2, \ldots, v_N\}$ is the set of nodes and $\mathcal{E} \subseteq \mathcal{V} \times \mathcal{V}$ is the set of edges. We denote the adjacency matrix as $\mathbf{A} \in \{0, 1\}^{N \times N}$, where $A_{i,j} = 1$ indicates nodes $v_i$ and $v_j$ are connected. We denote the node feature matrix as $\mathbf{X} \in \mathbb{R}^{N \times F}$, where $F$ is the number of features and $\mathbf{x}_i$ (the $i$-th row of $\mathbf{X}$) indicates the feature vector of node $v_i$. We denote the node labels for classification as $\mathbf{y} \in \{1, \ldots, M\}^N$, where $M$

is the number of classes. We denote the set of graph communities as $\mathcal{C} = \{C_1, C_2, \ldots, C_{N_c}\}$ where $N_c$ is the number of communities and each community $C_k$ is defined by a set of nodes $\mathcal{V}_{Ck}$ s.t. $\mathcal{V}_{Ci} \cap \mathcal{V}_{Cj} = \varnothing, \forall i, j \in \{1, 2, \ldots, N_c\}$ and $i \neq j$. We denote the subgraph containing the community $C_k$ as $G_{Ck} = (\mathcal{V}_{Ck}, \mathcal{E}_{Ck})$, where $\mathcal{E}_{Ck} \subseteq \mathcal{V}_{Ck} \times \mathcal{V}_{Ck}$ is the set of edges within this subgraph. With a bit of notation abuse, we use the union symbol to denote the combination of subgraphs, i.e., $G = \bigcup_{k=1}^{N_c} G_{Ck}$.

**Graph Neural Networks** Without the loss of generality, we take the commonly used graph convolutional network (GCN) [7] as an example when explaining GNNs in the following sections. The graph convolution operation of each GCN layer is defined as $\mathbf{H}^{(l)} = \sigma(\tilde{\mathbf{D}}^{-\frac{1}{2}} \tilde{\mathbf{A}} \tilde{\mathbf{D}}^{-\frac{1}{2}} \mathbf{H}^{(l-1)} \mathbf{W}^{(l)})$, where $l$ is the layer index, $\tilde{\mathbf{A}} = \mathbf{A} + \mathbf{I}$ is the adjacency matrix with added self-loops, $\tilde{\mathbf{D}}$ is the diagonal degree matrix $\tilde{D}_{ii} = \sum_j \tilde{A}_{ij}$, $\mathbf{H}^{(0)} = \mathbf{X}$, $\mathbf{W}^{(l)}$ is the learnable weight matrix at the $l$-th layer, and $\sigma(\cdot)$ denotes a nonlinear activation such as ReLU.

**Graph Data Augmentation** We follow prior literature [13] to classified graph data augmentation methods for node classification into two categories: stochastic operations for original-graph setting and deterministic operations for modified-graph setting. Let $h : G \to G_m$ be a graph data augmentation operation that generates a variant $G_m$ of the original graph $G$. In the original-graph setting, $h$ can be stochastic and applying it for $T$ times results with $T$ graph variants $G_m$, such that $G \cup \{G_m^i\}_{i=1}^T$ is used in training while only $G$ is used for inference. On the other hand, in the modified-graph setting, $h$ is deterministic and outputs one $G_m$, such that $G_m$ replaces $G$ for both training and inference.

In this work, we consider four typical state-of-the-art graph data augmentation operations for node classification: $\mathcal{A} = \{\text{DROPEDGE, ATTRMASK, GAUGM\_ADD, GAUGM\_RM}\}^2$, which include both stochastic and deterministic operations and they apply on both graph structure and the node features. It's worth noting that our proposed AutoGDA is not limited to these four augmentation operations and can take any graph data augmentation operations in $\mathcal{A}$.

DROPEDGE [18] and ATTRMASK [26] are stochastic augmentation operations, where they randomly drop (mask) a given percentage of edges (node attributes) in each training epoch of GNN. DROPEDGE implies the graph structure has certain robustness to the edge connectivity and also alleviates the well-known over-smoothing problem of GNNs [18]. ATTRMASK encourages GNNs to recover masked node attributes with their context information in the local neighborhood. On the other hand, GAUGM\_ADD and GAUGM\_RM [13] are deterministic augmentation operations that deterministically modify the graph structure to promote the graph's homophily and hence improve the model's performance for node classification [13]. GAUGM\_ADD, and GAUGM\_RM use the predicted edge probabilities for all node pairs by VGAE [28] and add new edges (remove existing edges) with highest (lowest) edge probabilities.

Each of the above four augmentation operations has one parameter controlling the percentage magnitude of the augmentation, resulting with a $100^4$ searching space when finding the optimal strategy for each dataset. As we separate the graph into multiple communities and aim to search for the optimal community customized augmentation strategy, the searching space would be $100^{4 \times N_c}$, where $N_c$ is the number of communities. As $N_c$ gets larger, the searching space becomes infeasible for traditional parameter searching methods such as grid search. Therefore, a new efficient approach for automated graph augmentation search is desired.

**Problem Definition** Following the definition of previous literature [13, 18] on graph data augmentation for node classification, the problem of finding a hand-crafted graph data augmentation strategy can be defined as follows. Given the graph data $G$, a set of graph data augmentation operations $\mathcal{A}$, and GNN model $f : G \to \hat{\mathbf{y}}$, find the best strategy of applying $\mathcal{A}$ on $G$ such that the node classification performance of $f$ on $G$ is maximized.

In this work, as we propose to find the best augmentation strategy for each community in the graph, the problem of automated graph data augmentation is then defined as: **given** the graph data $G$, graph communities $\mathcal{C}$ in $G$ (the community detection method for generating $\mathcal{C}$ is treated as a hyperparameter), a set of graph data augmentation operations $\mathcal{A}$, and GNN model $f : G \to \hat{\mathbf{y}}$, **find** the best strategy of applying $\mathcal{A}$ on the subgraphs of each community $C_k \in \mathcal{C}$ such that the node classification performance

---

$^2$To differentiate from the baseline methods (normal font), we use the small caps font to denote the augmentation operations.

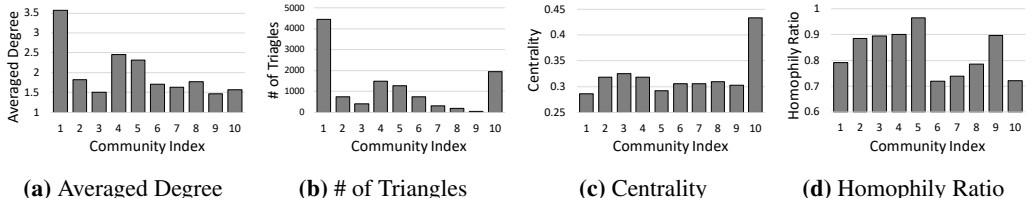

**(a)** Averaged Degree    **(b)** # of Triangles    **(c)** Centrality    **(d)** Homophily Ratio

**Figure 1:** Graph communities detected by the Louvain method on the PubMed dataset show diverse distribution on different characteristics of graph structure.

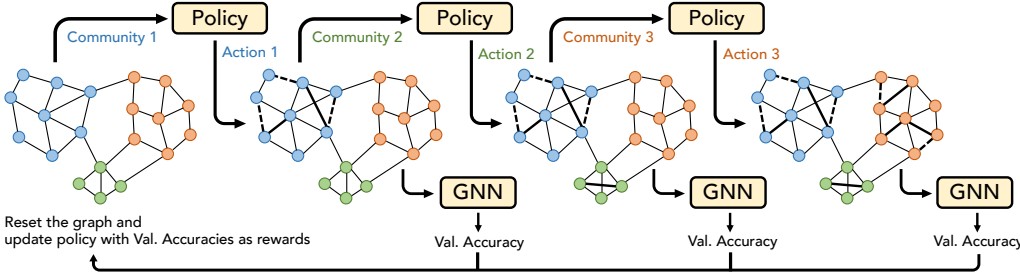

**Figure 2:** Overview of one iteration of our proposed AutoGDA on an example graph with three communities. In each step, the policy network takes the observation of one graph community as input and outputs the augmentation strategy for it. The GNN is then fine-tuned with the augmented graph and the validation accuracy is used as the reward to update the policy network.

of $f$ on $G$ is maximized. The main difference between our problem definition and the previous literature is the use of graph communities $\mathcal{C}$ to find the best data augmentation strategies.

## 3    Automated Graph Data Augmentation

Section 3.1 shows our motivation of customizing graph data augmentation strategies for different communities. Section 3.2 models automated graph data augmentation as a bilevel optimization problem. Section 3.3 presents the reinforcement learning-based framework AutoGDA.

### 3.1    Motivation

Community structures naturally exist in graphs [19, 20] and graph community detection has been extensively studied in the past few decades [29]. Graph community detection methods (e.g., the Louvain method [25]) separates the set of nodes in the graph into disjoint subsets such that the quality of the communities, which is usually measured by modularity [20], are maximized. Thus, the nodes within the same community are more densely connected and also more important to each other for node classification [21, 30] comparing with the nodes in different communities.

Our idea is based on the observation that different communities in the same graph mostly show disparate data distribution, which was also shown in previous literature [22–24]. The structure of communities commonly varies in terms of density, centrality, etc. For example, Figure 1 shows the characteristics of communities (detected by Louvain) on the PubMed graph [7]. Figures 1(a), 1(b), and 1(c) show the distributions of averaged degree, number of triangles, and centrality, respectively. Figure 1d presents the homophily ratio [31] of community subgraphs, where the homophily ratio is calculated by the fraction of edges which connect nodes that have the same class label. Previous works [11, 13] have shown that the graph homophily is strongly correlated with node classification performance of GNNs, because semi-supervised graph learning methods are mainly based on the homophily assumption [32]. From Figure 1 we observe that the communities disparate with different distributions under different measurements.

Moreover, for certain deterministic graph data augmentation methods such as GAugM [13]. The minority communities may be ignored during the augmentation process. For example, GAugM [13] modifies the graph structure according to the edge probabilities given an trained edge predictor, in which it adds missing edges with highest probabilities and remove existing edges with lowest

probabilities. However, it is possible that all of the modification would happen in only one or few graph communities, which shows the strongest homophily patterns.

With the above observations on the disparate characteristics of graph communities, we argue that the state-of-the-art methods that apply the augmentation operations on the whole graph may not be the best practice of graph data augmentation. Auto-augmentation literature in computer vision [2, 5] has shown that customizing augmentation operations for data objects/batches is more effective than using the same strategy for the entire dataset. Although it is infeasible to learn the best operation for each data object (node) for node classification due to the dependent nature of graph data, graph data augmentation could benefit from having customized augmentation strategy for each community. The next sub-section formulates the problem of automated graph data augmentation via community customization.

## 3.2 Bilevel Optimization Formulation

We formulate the problem of automated graph data augmentation in a similar way to the auto-augmentation problem in vision tasks [2, 5]: it aims to find a set of graph data augmentation operations for each community in the graph, which maximizes the performance of a graph neural network model on the task of (semi-)supervised node classification.

Let the graph data augmentation policy network be defined as $g_\theta : G \to \{0, 1, \dots, 99\}^{|\mathcal{A}|}$, which is a multi-layer perceptron (MLP) that is parameteraized by $\theta$. The policy takes a (sub)graph as input and outputs the augmentation strategy for this (sub)graph, which in our case is the four percentage magnitudes for $\mathcal{A} = \{\text{DROPEDGE}, \text{ATTRMASK}, \text{GAUGM\_ADD}, \text{GAUGM\_RM}\}$. Let $aug(g_\theta(G))$ be a function that applies the augmentation strategy $g_\theta(G)$ on (sub)graph $G$, then the automated graph data augmentation process for subgraph $G_{Ck}$ is formulated as

$$G'_{Ck} = aug\big(g_\theta(G_{Ck})\big) \tag{1}$$

where $G'_{Ck}$ denotes the subgraph $G_{Ck}$ after augmentation.

We denote the GNN model as $f_\omega : G \to \hat{\mathbf{y}}$, which is parameteraized by $\omega$. It takes the graph as input and outputs the predicted node labels $\hat{\mathbf{y}}$. Let the $\mathbf{y}_{tr}$ and $\mathbf{y}_{val}$ be the node labels for training set and validation set, respectively. The objective of obtaining the best augmentation policy (solving for $\theta$) could be described as a bilevel optimization problem [33]. The inner level is the model weight optimization, which is solving for the optimal $\omega_\theta$ given a fixed augmentation policy ($g_\theta$):

$$\omega_\theta = \arg\min_\omega \mathcal{L}\bigg( f_\omega\Big( \bigcup_{k=1}^{N_c} aug\big(g_\theta(G_{Ck})\big)\Big), \mathbf{y}_{tr} \bigg), \tag{2}$$

where $\mathcal{L}$ denotes the loss function (cross entropy).

The outer level is the augmentation policy optimization, which is optimizing the policy parameter $\theta$ using the result of the inner level problem. Here we take the validation performance (accuracy) as the optimization objective. Then we have the problem formulated as below:

$$\theta^* = \arg\max_\theta ACC\bigg( f_{\omega_\theta}\Big( \bigcup_{k=1}^{N_c} aug\big(g_\theta(G_{Ck})\big)\Big), \mathbf{y}_{val} \bigg), \\ \text{where } \omega_\theta = \arg\min_\omega \mathcal{L}(\omega, \theta) \text{ (Eq. (2))}, \tag{3}$$

where $\theta^*$ denotes the parameter of the optimal policy, and $ACC(f(G), \mathbf{y}_{val})$ denotes the validation accuracy.

## 3.3 AutoGDA Framework

As the graph data augmentation operations $\mathcal{A}$ modifies the graph structure and also affects the training of the GNN model, when applying the augmentation operations community by community, the graph data augmentation can be formulated as a sequential process on the graph. Figure 2 illustrates the sequential process: in each step of the iteration, the policy network takes the observation of a community and outputs the action containing the set of augmentation magnitudes which will be applied on this community. We finetune the GNN model after applying the augmentations and take the validation accuracy as rewards.

**Parameter Sharing** As the solving of bilevel optimization problems is extremely time consuming due the repeated solving of the inner loop [33], we utilize the weight sharing scheme for automated augmentation proposed by Tian et al. [5]. At the start of each episode, we reset the current graph to the original graph, pretrain the GNN model on the original graph (without optimizing the outer loop or applying any data augmentation operation), and obtain $\bar{\omega}$. That is,

$$\bar{\omega} = \arg\min_{\omega} \mathcal{L}(f_{\omega}(G), \mathbf{y}_{tr}) \tag{4}$$

In each step of this episode, instead of training a new GNN model from scratch to get $\omega_{\theta}$, we load the parameters $\bar{\omega}$ from pretraining and finetune the GNN model for only a small number of epochs with the given actions (augmentation strategy) to get $\bar{\omega}_{\theta}$. Therefore, the outer level for optimizing the augmentation policy parameters becomes

$$\theta^* = \arg\max_{\theta} ACC\bigg( f_{\bar{\omega}_{\theta}}\Big( \bigcup_{k=1}^{N_c} aug\big(g_{\theta}(G_{Ck})\big)\Big), \mathbf{y}_{val} \bigg). \tag{5}$$

**Reinforcement Learning (RL) Environment** The set of graph data augmentation operations $\mathcal{A}$ contains both stochastic and deterministic augmentation operations, where the stochastic operations (DROPEDGE, ATTRMASK) affect the GNN model in training and the deterministic operations (GAUGM_ADD, GAUGM_RM) directly modifies the graph structure. Therefore, as we apply the augmentation strategy on each community, the node/graph representation obtained by GNN trained with the augmentations also changes. This process forms a Markov decision process, whose length is equal to the number of communities.

In the RL environment, we take the current graph with the given augmentation strategy as the state and use the graph representation of one community as the observation for one step. That is, for each graph community $C_k$, the observation is the pooled graph representation of the subgraph $G_{Ck}$, i.e., the element-wise mean of the node representations for all nodes in $\mathcal{V}_{Ck}$. As the output dimension of GNN's last layer is the number of unique labels, which is usually very small, we take the output of the GNN's second last layer as the node representations. The policy network $g_{\theta}$ takes the observation (i.e., graph representation of the community) as input and outputs the magnitudes of different augmentation operations for the community. Note that the augmentation operation would not be applied if its magnitude is zero.

For optimizing our proposed method, we opt for simplicity and employ the widely-used Proximal Policy Optimization (PPO) [34] algorithm. We use the validation performance of the GNN model after finetuning in step as the reward to the RL policy.

**Summary** Algorithm 1 summarizes the whole process of AutoGDA. In each episode of the policy optimization stage, we first pretrain the GNN model by Eq. (4) and reset the graph. The subgraphs $G'_{Ck}$ in line 3 are for tracking the augmented communities. Then for each community, we first obtain and apply its augmentations strategy given by the policy (line 5), then load the pretrained parameters $\bar{\omega}$ for the GNN model, finetune it with the current augmentation strategies, and use the validation accuracy as reward to update the policy. After the policy network is sufficiently trained, we get the final graph data augmentation strategy for the whole graph (all communities) by the trained policy $g_{\theta^*}$. Finally, we reset GNN and train it with $\bigcup_{k=1}^{N_c} aug(g_{\theta^*}(G_{Ck}))$ to get the predicted labels $\hat{\mathbf{y}}_{test}$.

---

**Algorithm 1:** AutoGDA

**Input** : $G, \mathcal{C}, \mathbf{y}_{tr}, \mathbf{y}_{val}$
/* Policy Optimization */
1 **for** *episode in range(n_episodes)* **do**
2     Pretrain GNN and obtain $\bar{\omega}$ by Eq. (4) ;
3     $\{G'_{C1}, \ldots, G'_{CN_c}\} = \{G_{C1}, \ldots, G_{CN_c}\}$ ;
4     **for** *k in* $\{1, 2, \ldots, N_c\}$ **do**
5        $G'_{Ck} = aug(g_{\theta}(G_{Ck}))$ ;    // Eq. (1)
6        Load $\bar{\omega}$ ;
7        Finetune GNN with $\bigcup_{k=1}^{N_c} G'_{Ck}$ and get $\bar{\omega}_{\theta}$ ;
8        Use val. ACC to update $\theta$ ;    // Eq. (5)
9     **end**
10 **end**
11 $\theta^* = \theta$ ;
/* Inference */
12 $G^* = \bigcup_{k=1}^{N_c} aug(g_{\theta^*}(G_{Ck}))$ ;
13 Reset and train GNN with $G^*$ and get $\hat{\mathbf{y}}_{test}$ ;
**Output :** $\hat{\mathbf{y}}_{test}$

---

**Table 1:** Summary statistics and experimental setup for the datasets.

|                     | Cora  | CiteSeer | PubMed | Flickr  | ECom20k | ECom43k |
|---------------------|-------|----------|--------|---------|---------|---------|
| # Nodes             | 2,708 | 3,327    | 19,717 | 7,575   | 20,799  | 43,117  |
| # Edges             | 5,278 | 4,552    | 44,338 | 239,738 | 47,661  | 117,469 |
| # Features          | 1,433 | 3,703    | 500    | 12,047  | 132     | 132     |
| # Classes           | 7     | 6        | 3      | 9       | 2       | 2       |
| # Training nodes    | 140   | 120      | 60     | 757     | 2,275   | 4,209   |
| # Validation nodes  | 500   | 500      | 500    | 1,515   | 2,275   | 4,209   |
| # Test nodes        | 1,000 | 1,000    | 1,000  | 5,303   | 6,825   | 12,627  |

**Table 2:** Node classification accuracy across GNN architectures and public benchmarks. Bolded are the best performance and the comparable ones (within the standard deviation of the best performance).

| GNN   | Method    | Cora              | CiteSeer          | PubMed            | Flickr            |
|-------|-----------|-------------------|-------------------|-------------------|-------------------|
| GCN   | Original  | 81.5±0.4          | 70.3±0.5          | 79.0±0.3          | 61.2±0.4          |
|       | +AdaEdge  | 81.9±0.7          | **72.8**±0.7      | 79.8±0.4          | 61.2±0.5          |
|       | +DropEdge | 82.0±0.8          | 71.8±0.2          | 79.3±0.3          | 61.4±0.7          |
|       | +FLAG     | 80.2±0.3          | 68.1±0.5          | 78.5±0.2          | 62.3±0.4          |
|       | +GAugM    | 83.5±0.4          | 72.3±0.4          | 80.2±0.3          | 68.2±0.7          |
|       | +GAugO    | 83.6±0.5          | **73.3**±1.1      | 79.3±0.4          | 62.2±0.3          |
|       | +AutoGDA  | **84.4**±0.3      | **73.0**±0.4      | **81.6**±0.5      | **71.4**±0.5      |
| GSAGE | Original  | 81.3±0.5          | 70.6±0.5          | 78.3±0.6          | 57.4±0.5          |
|       | +AdaEdge  | 81.5±0.6          | 71.3±0.8          | 78.5±0.2          | 57.7±0.7          |
|       | +DropEdge | 81.6±0.5          | 70.8±0.5          | 78.7±0.7          | 58.4±0.7          |
|       | +FLAG     | 79.2±0.9          | 67.9±1.4          | 77.4±0.3          | 48.5±0.6          |
|       | +GAugM    | **83.2**±0.4      | 71.2±0.4          | 78.7±0.3          | 65.2±0.4          |
|       | +GAugO    | 82.0±0.5          | **72.7**±0.7      | 79.4±0.9          | 56.3±0.6          |
|       | +AutoGDA  | **83.2**±0.5      | **72.5**±0.4      | **80.0**±0.5      | **73.4**±0.6      |
| GAT   | Original  | 83.0±0.7          | 72.5±0.7          | 79.0±0.3          | 46.9±1.6          |
|       | +AdaEdge  | 82.0±0.6          | 71.1±0.8          | 79.1±0.6          | 48.2±1.0          |
|       | +DropEdge | 81.9±0.6          | 71.0±0.5          | 78.9±0.6          | 50.0±1.6          |
|       | +FLAG     | 79.6±0.6          | 67.7±0.7          | 78.2±0.5          | 48.9±1.1          |
|       | +GAugM    | 82.1±1.0          | 71.5±0.5          | 79.0±0.5          | 63.7±0.9          |
|       | +GAugO    | 82.2±0.8          | 71.6±1.1          | 78.5±0.8          | 51.9±0.5          |
|       | +AutoGDA  | **84.8**±0.2      | **73.2**±0.4      | **79.8**±0.6      | **65.1**±0.9      |

For the edges which connect nodes that belong to different communities, we make them an optional special community of edges ($C_{N_c+1}$) in AutoGDA. In the case we use this community, it only disjoint with others in edges (i.e., $\mathcal{V}_{CN_c+1} \cap (\cup_{k=1}^{N_c} \mathcal{V}_{Ck}) = \varnothing$ and $\mathcal{E}_{CN_c+1} = \mathcal{E} \backslash (\cup_{k=1}^{N_c} \mathcal{E}_{Ck})$), so we only apply DROPEDGE and GAUG_RM on it as the other two augmentation operations are covered by other communities.

## 4 Experiments

In this section, we evaluate the performance of the proposed AutoGDA across different GNN backbones and datasets, and over alternative graph data augmentation methods.

### 4.1 Experimental Setup

**Datasets** We evaluate with 4 public benchmark datasets across domains: citation networks with strong homophily (Cora, CiteSeer, PubMed [7]) and social networks that exhibits heterophily [31] (Flickr [35]). We also evaluate with 2 real industry application benchmarks ECom20K and ECom43K for the task of graph anomaly detection. The statistics of all datasets are provided in Table 1.

**Baselines** We evaluate the AutoGDA and baselines using 3 widely used GNN architectures: GCN [7], GraphSAGE [8], and GAT [36]. We compare the node classification performance of AutoGDA with that achieved by standard GNN, as well as four state-of-the-art graph data augmentation baselines: AdaEdge [11], DropEdge [18], GAugM [13], and GAugO [13]. To evaluate the robustness of

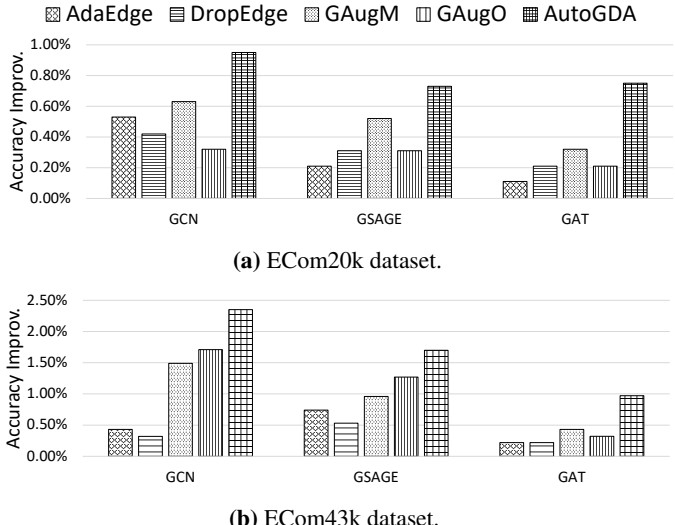

**(a)** ECom20k dataset.

**(b)** ECom43k dataset.

**Figure 3:** Relative improvements over GNNs for accuracy on two real-world industry anomaly detection datasets.

**Table 3:** Node classification accuracy against different levels of adversarial attacks. Bolded are the best performance and the comparable ones (within the standard deviation of the best performance).

| Dataset | Attack Method Ptb. Rate | DICE [40] | | | Metattack [27] | | |
|---|---|---|---|---|---|---|---|
| | | 10% | 30% | 50% | 10% | 30% | 50% |
| Cora | GCN [7] | 78.4±0.6 | 73.6±0.9 | 66.8±1.2 | 70.2±0.9 | 32.6±2.0 | 16.6±0.8 |
| | GAugO [13] | 78.8±0.4 | 74.0±0.3 | **67.3**±0.5 | 72.5±1.1 | 57.1±0.7 | 40.6±1.1 |
| | GNN-Jaccard [38] | 73.1±0.5 | 66.4±0.5 | 66.9±0.4 | 72.1±0.7 | 51.6±0.5 | 38.4±0.7 |
| | ElasticGNN [39] | **79.8**±0.8 | **74.5**±0.8 | **67.6**±1.4 | **74.3**±1.2 | 49.0±1.3 | 35.4±1.4 |
| | AutoGDA | **80.2**±0.7 | **74.7**±0.3 | **67.9**±0.9 | **74.5**±1.0 | **63.2**±1.9 | **53.7**±1.2 |
| CiteSeer | GCN [7] | 65.8±1.1 | 60.1±0.8 | 56.5±0.8 | 38.4±1.0 | 15.9±0.8 | 10.7±1.9 |
| | GAugO [13] | **67.0**±0.5 | **60.9**±0.4 | 56.5±0.9 | 50.2±0.9 | 34.3±0.6 | 29.1±1.2 |
| | GNN-Jaccard [38] | 66.5±1.3 | 59.8±0.7 | 54.1±1.0 | 43.7±1.2 | 27.0±0.7 | 19.8±0.4 |
| | ElasticGNN [39] | 66.7±1.4 | 59.7±0.8 | 56.3±1.4 | 47.5±1.3 | 31.8±0.3 | 23.5±4.3 |
| | AutoGDA | **67.8**±1.1 | **62.1**±1.3 | **57.6**±1.1 | **61.5**±1.6 | **52.6**±1.3 | **47.8**±1.7 |
| PubMed | GCN [7] | 73.9±0.3 | 67.1±0.3 | 63.6±0.7 | 67.2±0.4 | 41.3±1.5 | 27.5±1.7 |
| | GAugO [13] | 74.6±0.4 | 67.8±0.3 | 64.8±0.5 | 71.6±0.9 | 51.8±0.7 | **40.7**±1.0 |
| | GNN-Jaccard [38] | 73.7±0.4 | 67.3±0.5 | 64.2±0.4 | 68.2±0.4 | 41.9±0.9 | 29.1±1.1 |
| | ElasticGNN [39] | **75.5**±0.6 | **68.7**±0.7 | **65.4**±0.7 | 71.6±0.5 | 49.8±0.4 | 40.1±0.8 |
| | AutoGDA | 75.1±0.8 | 68.6±0.5 | **65.6**±0.4 | **73.1**±1.2 | **55.9**±1.8 | **42.2**±1.6 |

AutoGDA under graph adversarial attacks [37], we evaluate AutoGDA and state-of-the-art graph defensing methods (GNN-Jaccard [38] and ElasticGNN [39]) against graph adversarial attacks: DICE [40] and Metattack [27].

## 4.2 Experimental Results

We show comparative results of AutoGDA and baseline methods in Tables 2 and 3 and Figure 3. Table 2 is organized per GNN architecture (row), per dataset (column), and different methods (within-row). Table 3 is organized per dataset (row), per graph adversarial attack method (column), per attack level (within-column), and different methods (within-row). For customer privacy concern, relative improvements instead of the performances are reported in Figure 3. We bold the best and comparable performances. In short, our proposed AutoGDA consistently achieve the best or comparable performances in all combinations of GNN architectures and datasets.

**Effectiveness on graph data augmentation** From Table 2 and Figure 3 we observe that our proposed AutoGDA achieves improvements over all three GNN architectures (averaged across datasets):

**Table 4:** Ablation experiments using GCN on PubMed.

|  | PubMed |
| --- | --- |
| DropEdge | 79.3±0.3 |
| AutoGDA with DROPEDGE (single community) | 79.3±0.4 |
| AutoGDA with DROPEDGE | 79.4±0.2 |
| GAugM | 80.2±0.3 |
| AutoGDA with GAUGM (single community) | 80.2±0.3 |
| AutoGDA with GAUGM | 81.2±0.4 |
| AutoGDA (single community) | 80.4±0.3 |
| AutoGDA | **81.6**±0.5 |

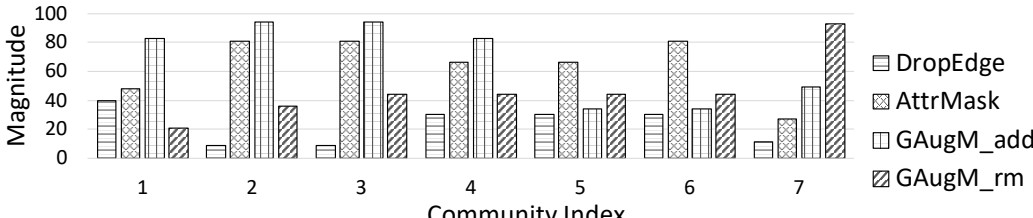

**Figure 4:** AutoGDA learns diverse augmentation strategies for different communities in the PubMed graph.

AutoGDA improves 5.0% (GCN), 6.1% (GraphSAGE), and 7.5% (GAT), repsectively. From the dataset perspective, AutoGDA also achieves improvements over all 6 datasets (averaged across GNN architectures): AutoGDA improves 2.7%, 2.2%, 2.1%, 27.6%, 0.8%, and 1.7% respectively for each dataset (Cora, CiteSeer, PubMed, Flickr, ECom20k, ECom43k). Notably, AutoGDA is especially effective on social networks (Flickr), which are naturally more heterophily. Although GAugM [13] outperformed all other baselines with large margin, AutoGDA still significantly improved from GAugM by combining the advantages of different augmentations and learning the best combined strategy. Finally, we note that AutoGDA outperforms all graph data augmentation baselines: specifically, AutoGDA improves 5.6%, 5.4%, 6.1% 2.0%, and 4.7% respectively over AdaEdge, DropEdge, FLAG, GAugM, and GAugO (averaged across datasets and GNNs). We reason that learning customized augmentation for each graph community and combining several state-of-the-art graph data augmentation operations both contributed to the performance improvement of AutoGDA.

**Robustness against graph adversarial attacks** From Table 3 we observe the proposed AutoGDA is able to effectively learn robust representation under graph adversarial attacks. Although the recently baseline ElasticGNN [39] also achieved good performance against Random Injection and DICE [40], AutoGDA outperformed ElasticGNN with large margins under Metattack [27]. In short, our proposed AutoGDA achieved the best performance for 21 our of 27 combinations of datasets and graph adversarial attack methods.

**Necessity of community adaptive augmentations** Table 4 shows ablation experiments on PubMed using GCN. We compare the performances of AutoGDA using only one augmentation operations versus using all augmentation operations in $\mathcal{A}$, and we also compare the performances of AutoGDA with single community (viewing the whole graph as the only community) versus our default setting (using Louvain method for community detection). We note that when using only one augmentation operation (combining GAUGM_ADD and GAUGM_RM as GAUGM) and under single community setting, AutoGDA performs parameter search on the existing graph data augmentation methods. We also observe from Table 4 that the default AutoGDA (with multiple graph communities) consistently outperform the single community setting, showing the community customized augmentation strategy is crucial for the performance improvement.

**Case Study** Figure 4 showcases the learned augmentation strategies for seven different communities on PubMed dataset by our proposed AutoGDA with GCN. We observe that our propose AutoGDA is able to learn diverse augmentations strategies for different communities in the graph.

# 5 Related Work

**Graph Neural Networks** GNNs enjoy widespread use in modern graph-based machine learning due to their flexibility to incorporate node features, custom aggregations and inductive operation, unlike earlier works which were based on embedding lookups [41–43]. In recent years, many spectral GNN variants [7, 44–48] were proposed following the initial idea of convolution based on spectral graph theory [49]. As spectral GNNs usually requires (expensive) operations on the full adjacency matrix, spatial GNNs which perform graph convolution with neighborhood aggregation became prominent [8, 36, 50–52] owing to their scalability and flexibility [53]. Several other works also proposed advanced architectures which add residual connections to facilitate deep GNN training [54, 55]. GraphMix [56] proposed to regularize the GNN model with a fully connected network. More recently, graph neural architecture search (NAS) methods [57, 58] utilizing reinforcement learning were proposed to learn the optimal GNN architecture.

**Graph Data Augmentation** As GNNs have emerged as a rising approach for learning with graph data, Graph Data Augmentation (GDA) [17, 59–62] for GNNs were proposed and studied in recent years. Due to the complex, non-Euclidean structure of graphs, most GDA work focused on manipulating the graph structure [13, 63, 63]. DropEdge [18] randomly drops a fraction of edges during each training epoch, in a way similar to Dropout [64]. Following DropEdge, several works [65–67] proposed methods that learns to drop instead of dropping at random. Graph structure learning methods [46, 68–71] can also be viewed as graph data augmentation as they learn from graphs whose structures are partially or totally unknown. AdaEdge [11] and BGCN [72] are iterative methods that updates the graph structure with the prediction of GNNs. Zhao et al. [13] showed that graph homophily critically affects message passing-based GNNs and proposed GAugM and GAugO that manipulate the graph structure with the edge probabilities given by VGAE [28] to augment the graph data. Kong et al. [12] and Tang et al. [15] proposed augmentation methods that operates on the node features. Graph structual learning methods [70, 73, 74] search for better graph structure that augments the initial graph structure, with the goal of optimizing the graph for downstream tasks. Aside from the methods that directly use GDA for semi-supervised node classification, several works that used GDA in self-supervised graph learning were also proposed. NodeAug [75] and Grand [32] used augmentation with self-supervised consistency loss as an additional term to the cross entropy loss. GraphCL [26] used augmentation in self-supervised graph contrastive learning for graph classification. Eland [16] proposed action sequence augmentation for graph anomaly detection. Zhao et al. [76] studied counterfactual data augmentation for link prediction.

**Automated Data Augmentation** Several automated data augmentation approaches [2–4, 6, 77] have been proposed in CV in the past few years. These methods seek to find the optimal data augmentation policies for each given dataset automatically. Cubuk et al. [2] formulated the automated data augmentation problem as a discrete search problem and proposed a reinforcement learning framework to search the best augmentation operations via proxy tasks (i.e., a smaller model). Several works [3, 4, 6] were then proposed to improve the efficiency of automated data augmentation. Cubuk et al. [78] showed that the models with different number of parameters benefits from different magnitude of augmentation operations and proposed RandAug that searches for one augmentation magnitude that is used for all operations. Tian et al. [5] proposed the Augmentation-Wise Weight Sharing method that enables a fast evaluation process on the original model while not sacrificing the efficiency. Recently, several automated augmentation methods [79–81] have been proposed for self-supervised graph representation learning. Luo et al. [82] studied automated data augmentation for the task of graph classification. Nevertheless, automated data augmentation is rather under-explored for (semi-)supervised node classification tasks.

# 6 Conclusions

In this paper, we studied the problem of community-customized data augmentation for node classification. Our work showed that different communities require different augmentation strategies for the best node classification performance due to their disparate characteristics. We proposed an automated graph data augmentation framework AutoGDA that adopted reinforcement learning to automatically learn the optimal augmentation strategy for each community. Through extensive experiments on benchmark graph datasets from multiple domains, our proposed approach AutoGDA achieved up to 12.5% accuracy improvement over the state-of-the-art graph data augmentation baselines.

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

# A  Additional Dataset Details

In this section, we provide some additional, relevant dataset details.

**Citation networks.**  Cora, CiteSeer and PubMed are citation networks that are commonly used as benchmarks in GNN-related prior works [7, 11, 13, 18, 36]. In these citation networks, the nodes are published papers; the features are preprocessed (e.g., bag-of-words) vectors of the corresponding paper title and/or abstract; the edges represent the citation relation between papers; the labels are the category of each paper.

**Social networks.**  Flickr is an online social network platform, where users can also follow each other as well as posting images and videos. The user-specified list of interest tags are used as user features and the groups that users joined are used as labels [35].

**E-commerce networks.**  ECom20k and ECom43k are e-commerce networks that were constructed with customer purchase/review records from a leading international e-commerce website.  The network contains four types of nodes: customers, sellers, products, and reviews, in which customer purchases products from sellers and leave reviews to products. They are two graphs constructed with records in different time periods. The task for these two datasets is abusive customer detection and the customers with golden labels are split into train/validation/test sets for node classification. The node features contains original node attributes as well as the one-hot encoded vectors of the node types. As the golden labels are limited and severely biased, the datasets are sampled with snowball sampling [83] with labeled abusive customers to ensure the relative independence of the graph. The node attributes used is anonymized and do not contain any personally identifiable information.

**Validation Method.**  For Cora, Citeseer, PubMed, and Flickr, we follow the commonly used semi-supervised setting in most GNN literature [7, 13, 36]. For ECom20k and ECom43k, we use 20/20/60% for train/validation/test splitting.

# B  Implementation Details

All the experiments in this work were conducted on either an AWS EC2 P4 Instance[3] or a G4dn Instance[4]. The P4 instance is equipped with 48 Intel Cascade Lake processor cores (96 vCPUs), 1.1 TB of RAM, and 8 Nvidia A100 GPU cards (40 GB of RAM each). The G4dn instance is equipped with 48 Intel Cascade Lake vCPUs, 192 GB of RAM, and 4 Nvidia T4 GPU cards (16 GB of RAM each). Note that although the EC2 instances are equipped with multiple GPU cards, AutoGDA only need one GPU to run all the experiments.

We report test accuracy averaged over 20 runs along with respective standard deviations. For baseline methods with same datasets in their original paper, we directly use their reported performances – numbers reported in the original paper are more preferred than the reproduced results in other papers.

## B.1  Baseline methods

For the new reproduced results in this work, all original GNN architectures are implemented in DGL[5] with Adam optimizer. For a fair comparison, we use hidden size of 128 for all methods. For baseline methods, we implemented AdaEdge [11] and DropEdge [18] with PyTorch and DGL, and used the official code packages[6] from the authors for GAugM and GAugO [13]. The hyperparameters for all baseline methods are tuned according to with the same range as in the proposed AutoGDA, and the hyperparameters for GAugM and GAugO [13] are tuned with the script by their authors.

## B.2  AutoGDA variants

As the same as for the baselines, all GNNs in AutoGDA are implemented with DGL and we used hidden size of 128.  We use the public PPO [34] implementation in the stable-baselines3

---

[3]https://aws.amazon.com/ec2/instance-types/p4/
[4]https://aws.amazon.com/ec2/instance-types/g4/
[5]https://www.dgl.ai/
[6]https://github.com/zhao-tong/GAug

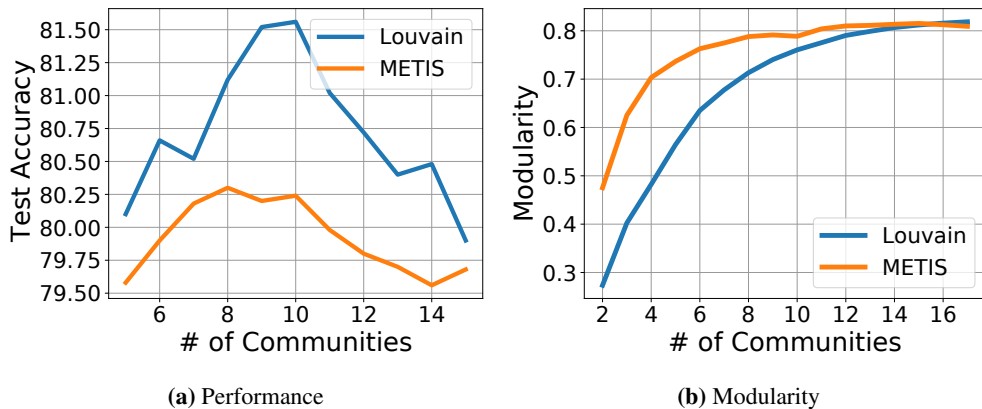

**(a)** Performance

**(b)** Modularity

**Figure 5:** AutoGDA is robust to the choices of community detection algorithms as well as the number of communities. The number of communities can be decided with the modularity measurement.

package[7] and implement our RL environment with gym[8]. All parameters for PPO algorithm are set as default from `stable-baselines3`. We use the Louvain method [25] as the default community detection method for all experiments in Section 4 except the ones in the sensitivity analysis (Figure 5). The number of communities is treated as a hyperparameter and determined with the help of modularity measurement as described in the following sensitivity analysis.

## C   Additional Experimental Results

**Sensitivity of AutoGDA .**   Figure 5a shows the sensitivity analysis of our proposed AutoGDA on the choices of community detection methods and the number of communities. Figure 5b shows the modularity measurement for the two community detection methods (Louvain [25] and METIS [84]) at different number of communities. We observe that AutoGDA is generally good when $8 \leq N_c \leq 10$ for the PubMed dataset, which is also where the modularity curve converge. Thus, a good number of communities for AutoGDA is generally easy to find with the help of the modularity measurement.

---

[7]https://github.com/DLR-RM/stable-baselines3
[8]https://github.com/openai/gym

