# OpenReview forum: "AutoGDA: Automated Graph Data Augmentation for Node Classification"
_logconference.io/LOG/2022/Conference — LoG 2022 Poster_

### Official Review · Reviewer_Z29o · 2022-10-17

**Overall Score:** 5
**Confidence:** 4

**Review:**

This paper focuses on automated graph augmentation methods for node classification tasks. Authors are motivated by customizing augmentation operations in computer vision. Since the observation that communities in the same graph can show disparate data distribution, community customization can benefit node classification tasks from the perspective of data augmentation. This paper proposes an AutoAugmentation method for graph data through a Bilevel optimization formulation and reinforcement learning optimization.

Strength:
● This paper is an investigation for the exploration of applying AutoAugment techniques to graph data.
● AutoAugment on graph data is research-worthy since graphs are a kind of non-Euclidian data.

Weakness:
● The motivation of using community is not clear.
● This paper doesn't give enough analysis of the complexity of the proposed method. The training time and memory used by this framework deserve are need to be analyzed empirically since it is a huge problem for AutoAugment techniques.
● Whether it is worthwhile to apply reinforcement learning to graphs which is a kind of non-Euclidian data for augmentation? Further discussion is required.
● In computer vision, it has been proven that models can benefit from the combinations of different methods. Could you please give more explanations why the performance of GCN declines when combining AutoGDA with existing graph augmentation methods?
● Why the results seem to be rather stable on the condition that louvain algorithm is applied to the graph for community detection? Why the louvain is used here?

---

### Official Review · Reviewer_XmDZ · 2022-10-20

**Overall Score:** 5
**Confidence:** 4

**Review:**

Summary
The paper proposes a novel augmentation strategy for graphs, which consider the local community structure of graphs. The proposed method is distinguished from existing augmentation approaches in that the local properties are considered rather than the entire graph. The main motivation is that graphs contain communities and as each community exhibit a distinguished property, the augmentation strategy should also consider the community structure. The proposed augmentation strategy is shown to be effective and also robust against adversarial attacks.

Strong point
- The proposed method is timely, and it tackles an important problem in graph representation learning.
- The paper is well-written and generally easy to follow.

Weak point
- Although it is a new application to graph data, the proposed method is an existing method in computer vision as mentioned by the authors, and there seems to be not much of a challenge when adopting the existing method on graphs.
- As the authors show analyses on the disparate characteristics of communities using benchmark datasets in Figure 1, I expected the proposed method to somehow incorporate the characteristics such as degree, num. triangles, centrality and homophily ratio. However, these statistics were shown as a motivation for considering the community structure itself, which in my opinion was not expected.

Question to authors
- Since the proposed augmentation strategy works in a sequential order, I suspect that there should be an issue with scalability, which could be a serious problem in practice. Could you comment on the scalability?

---

### Official Review · Reviewer_RJeZ · 2022-10-25

**Overall Score:** 6
**Confidence:** 5

**Review:**

This paper studies the graph structure learning (GSL) problem, and proposes a bilevel optimization framework AutoGDA. Despite the effectiveness of some small datasets, I would recommend rejection due to the poor discussion and comparison of the studied area.

Strong points

S1. The proposed method handles an important problem.

S2. The paper is generally clear to follow.

Weak points

W1. Lack of comprehensive discussions on literature. The authors completely ignore the literature on graph structure learning (GSL), which defines is the exact studied problem, to name a few: IDGL [Chen et al., 2020],  LDS [Franceschi et al., 2019] (which is also a bi-level optimized model). There is already a survey of this area, see [Zhu et al., 2021] for details.

W2. The motivation of using communities is less explained.

W3. The experiments are unconvincing. More GSL baselines should be compared. I'm also concerned about the efficiency and scalability of the method as the experiments are conducted on small datasets. Results on the OGB benchmark could be far more convincing.

W4. Confusing notation of 99 in line 185 $G \rightarrow\{0,1, \ldots, 99\}^{|\mathcal{A}|}$, is this a typo?

[Franceschi et al., 2019] Learning Discrete Structures for Graph Neural Networks.

[Chen et al., 2020] Iterative Deep Graph Learning for Graph Neural Networks: Better and Robust Node Embeddings

[Zhu et al., 2021] A Survey on Graph Structure Learning

---

### Official Review · Reviewer_WbCV · 2022-10-26

**Overall Score:** 10
**Confidence:** 3

**Review:**

####################
Summary:
The paper presents AutoGDA, a graph augmentation method specifically tailored to the communities emerging from the graph structure. Unlike previous approaches proposing augmentation strategies for the whole graph, the authors design and implement a technique to learn a community-customized augmentation technique. First, the problem is modeled as a bilevel optimization task. The inner problem is finding the best model's parameters to minimize the classification loss on the training set, while the outer problem measures the reward (classification accuracy) on the validation. Then, given the high complexity of exploring the complete combinations of approaches for all detected communities, the authors suggest a reinforcement learning solution to efficiently find the optimal augmentation strategy for each community. Experiments on datasets accounting for citation networks, social networks, and e-commerce validate the proposed approach against state-of-the-art augmentation strategies for three popular graph architectures. Furthermore, an analysis of the efficacy of AutoGDA in an attack/defense scenario shows the robustness of the proposed approach also in these settings. Finally, an ablation study is run to demonstrate the importance of providing an adaptive augmentation technique for each community against the single-community setting.

####################
Pros:
The proposed approach is novel compared to the related literature and paves the way for other possible applications and scenarios exploiting graph augmentation customized to communities.
The paper is well-presented and easy to follow. The narrative follows a coherent flow that (1) introduces the scenario and the limitations of current approaches, (2) sets up the background and preliminaries, (3) motivates and formulates the approach, and (4) evaluates it.
The proposed approach is well-placed in the related literature (see Appendix).
The evaluation setting is extensive, with datasets from different domains, multiple compared graph baselines, graph augmentation strategies, attack/defense techniques on graphs, and an ablation study.
The authors provide implementation and reproducibility details in the Appendix and the code to run the presented experiments.
In the Appendix, the authors report on the performance variation when comparing different community detection strategies, number of communities, and data augmentation strategies in the case of heterogeneous graphs.

####################
Cons:
To the best of my knowledge, I cannot think of any.

####################
Reasons for score:
All the positive points outlined above would lead me to give a strong acceptance to the paper. The work presents an exciting and novel approach, with other possible scenarios and applications in the graph learning domain (especially when communities with different characteristics emerge from the graph). The paper is well presented, motivated, and formalized, with an extended evaluation spanning multiple baselines, strategies for augmentation, and addressed issues (e.g., robustness against adversarial attacks). The experimental setting is also well supported by the implementation and reproducibility details where code is provided. To conclude, I would be open to discussing the comments from my review with the authors and the other reviewers.

---

### Official Review · Reviewer_6wm2 · 2022-10-27

**Overall Score:** 6
**Confidence:** 5

**Review:**

This paper studied the problem of automated graph data augmentation and proposed AutoGDA, a reinforcement learning (RL) based automated augmentation method to further boost the performance of graph neural networks on the node classification task. The proposed method is overall novel and interesting. The whole framework is easy to follow and technically sound. I vote weak-accept in light of good writing, well-motivated problem, and simple yet effective method, but less convincing experimental settings and empirical results.

Strength:

- The proposed method is well-motivated and conceptually simple.
- Well-written and accessible paper.
- Experimental results are relatively comprehensive and complete.
- Good generalization results on graph benchmark datasets.

Major concerns:

- Sec 3.1 demonstrates the characteristics of different graph communities, which motivates their work that automatically designs augmentations for different communities. My major concern is that the author didn't show the effect of different augmentations on different communities. This makes the authors' claim and the proposed method less convincing. The author may need more experiments to show the effectiveness of different data augmentation performed on different communities.
- There are some missing related work/baselines needed to be included, e.g., JOAO[1], which also targets automated graph data augmentation.
- The experimental results in Table 2 are not consistent with that from ElasticGNN and Metattack. Actually, GCN with 10% perturbations generated by Metattack would not drop to 38.4% on Citeseer. In addition, the experimental setting under adversarial attacks is not common where the Ptb. Rate is too large and makes the attack noticeable and impractical for real-world scenarios. The authors are suggested to follow Metattack which used 5%, 10%, 15%, etc.
- What are the settings of baseline methods, e.g., the dropping probability of DropEdge? Did the authors carefully tune the hyperparameters for baselines and report their best performance?

Followup-questions/areas for improving score:

- After data augmentation, how to preserve properties in the original network, such as degree power-law distribution, and communities? If it was not necessary to preserve the properties, then what information should be preserved from the original graph?
- Deeper analysis of the (more interesting, I think) experimental results would be nice. I would like to see some discussion on the significant performance gain of AutoGDA over baselines across all datasets particularly Flickr.
- Datasets with small scales are not sufficient to evaluate the performance of the proposed method. I would suggest the authors conduct additional experiments on OGB benchmarks[2].

Some minor suggestions:

- Summarize the **heterophily** for different datasets (particularly homogeneous ones).
- Provide some introductions on the used RL methods would make this paper self-contained.

Reference:

[1] Graph Contrastive Learning Automated. ICML 21.
[2] Open Graph Benchmark: Datasets for Machine Learning on Graphs. NeurIPS 20.

---

I'd like to update my score after reading the authors' responses and comments from other reviewers.

---

### Meta-Review · Area_Chair_stHh · 2022-11-16

**Confidence:** 3
**Recommendation:** Accept

**Meta Review:**

The authors propose AutoGDA, a reinforcement learning-driven graph data augmentation technique that learns a set of augmentation strategies for each community in isolation. The motivation, that different communities in the same graph are subject to significantly different properties, and thus require different treatment during augmentation, is clever and sound.

There is some variance in the reviews, but (most/all) reviewers acknowledge that this is a well-written paper introducing a novel idea. The main criticism of reviewers is the lack of evaluation on larger datasets (which I share). AutoGDA is inherently sequential and expensive to run (i.e. takes 74 min to train on PubMed), so it might not even be applicable on larger datasets. Unfortunately, authors mention that they cannot currently run more experiments. Available runtimes in Figure 6 still suggests that AutoGDA is comparable to related work.

Other criticisms have been mostly erased by the authors in the rebuttal (e.g., comparison to related work, concerns about the parameter settings, lack of motivation, incremental methodological contribution).

Due to the convincing rebuttal and the novelty of the given approach, I am slightly leaning towards accepting this work. I thank all reviewers in engaging with the authors.

---

### Decision · Program_Chairs · 2022-11-22

Accept (Poster)